# Early evaluation of the 'STOP SEPSIS!' WHO Global Maternal Sepsis Awareness Campaign implemented for healthcare providers in 46 low, middle and high-income countries

Vanessa Brizuela [1], Mercedes Bonet [1], Carla Lionela Trigo Romero [2], Edgardo Abalos [3], Adama Baguiya [4], Bukola Fawole [5], Marian Knight [6], Pisake Lumbiganon [7], Meilė Minkauskienė [8], Ashraf Nabhan [9], Nafissa Bique Osman [10], Zahida P Qureshi [11], João Paulo Souza [2], On behalf of the World Health Organization Global Maternal Sepsis Study Research Group

Dr Bukola Fawole was deceased in January 2019

For numbered affiliations see end of article.

**Correspondence to**
Dr Vanessa Brizuela;
brizuelav@who.int

## ABSTRACT

**Objective** To evaluate changes in awareness of maternal sepsis among healthcare providers resulting from the WHO Global Maternal Sepsis Study (GLOSS) awareness campaign.

**Design** Independent sample precampaign/postcampaign through online and paper-based surveys available for over 30 days before campaign roll-out (pre) and after study data collection (post). Descriptive statistics were used for campaign recognition and exposure, and odds ratio (OR) and percentage change were calculated for differences in awareness, adjusting for confounders using multivariate logistic regression.

**Setting and participants** Healthcare providers from 398 participating facilities in 46 low, middle and high-income countries.

**Intervention** An awareness campaign to accompany GLOSS launched 3 weeks prior to data collection and lasting the entire study period (28 November 2017 to 15 January 2018) and beyond.

**Main outcome measures** Campaign recognition and exposure, and changes in awareness.

**Results** A total of 2188 surveys were analysed: 1155 at baseline and 1033 at postcampaign. Most survey respondents found the campaign materials helpful (94%), that they helped increase awareness (90%) and that they helped motivate to act differently (88%). There were significant changes with regard to: not having heard of maternal sepsis (−63.4% change, pre-OR/post-OR 0.35, 95% CI 0.18 to 0.68) and perception of confidence in making the right decisions with regard to maternal sepsis identification and management (7.3% change, pre-OR/post-OR 1.44, 95% CI 1.01 to 2.06).

**Conclusions** Awareness raising campaigns can contribute to an increase in having heard of maternal sepsis and an increase in provider perception of confidence in making correct decisions. Offering the information to make accurate and timely decisions while promoting environments that enable self-confidence and

### Strengths and limitations of this study

► This study presents the results of an evaluation of a global awareness campaign which accompanied a research study on maternal infections and sepsis.

► This evaluation was a cost-effective, feasible way in which to assess campaign effectiveness among a varied and global population of healthcare providers.

► Our precampaign/postcampaign using anonymous surveys with no control group does not allow to discern the impact of the campaign alone or matching precampaign and postcampaign responses.

► Campaign fidelity was only assessed through healthcare provider self-report at postcampaign surveys.

► This evaluation was restricted to the duration of the study follow-up period limiting understanding of long-term impact.

support could improve maternal sepsis identification and management.

## INTRODUCTION

The global health community has recently drawn attention to the importance of sepsis and its toll on global mortality and morbidity.[1–3] In 2017, the World Health Assembly approved a resolution on sepsis to improve the prevention, diagnosis and management of sepsis.[4] With updates in 2017 and 2018, the Surviving Sepsis Campaign has been developing guidelines for management and recommended bundles of care for sepsis among adult populations, not specific to pregnant or recently pregnant women, since 2002.[5–7]

---

**Box 1  Actions and components for the *STOP SEPSIS! awareness campaign***

► *Select a campaign lead*. A campaign lead was selected to coordinate and assist with the development of the campaign strategy and execution, and evaluation plan at a global level. This person ensured execution of each of the steps, supported the communication company and the study country coordinators in the participating countries who interacted with the providers working in the participating facilities.

► *Agree on a budget to fund the campaign*. Funds were necessary to cover the costs of the campaign lead, the communication company and support to countries for printing of materials. The cost of this campaign was US$200 000.

► *Seek the assistance of health and media communication experts*. A communication company with expertise in global health was contracted to lead the design and development of the Global Maternal Sepsis Study (GLOSS) campaign concept and look.

► *Decide on the minimum set of materials and activities to be developed and implemented*. With input from people in the field who would be targeted through the campaign, the decision to have posters, infographics, press release and other presentation templates, social media messaging and a website was agreed on. In addition, a global congress was conceived in collaboration with partners from the Global Sepsis Alliance.

► *Develop campaign messaging, image and logo*. A main message, tagline and logo were designed with assistance from the communication company, content experts in maternal sepsis and country/regional coordinators for GLOSS.

► *Develop an evaluation plan*. Given the breadth and geographical extent of the campaign's target population an online survey was used to collect providers' knowledge, attitudes, practices at baseline and postcampaign, including additional measures of campaign recognition and exposure at postcampaign. Paper-based surveys were used on demand.

► *Support the printing and upkeep of materials*. The campaign lead coordinated translation of all materials into five United Nations (UN) official languages and three additional languages as per GLOSS country coordinators' request. Participating countries were provided with funds needed to print the posters and infographics. Campaign lead was also in charge of regular upkeep of the dedicated website which includes timely news stories.

► *Implement the campaign*. This included:
  – *World Sepsis Congress (WSC) Spotlight*. A free, online congress focusing specifically on maternal and neonatal sepsis offered in collaboration with the Global Sepsis Alliance (https://wscspotlight.org/). The 25 presentations given over four sessions were later made available as YouTube videos and podcasts for free, with subtitles in multiple languages.
  – *Website*. A dedicated website used both as a repository of campaign materials for free download and to disseminate news about the study (http://srhr.org/sepsis).
  – *Print materials*. Posters with information about the study and infographics on maternal sepsis prevention, and identification and management to be displayed in different areas where women with suspected or confirmed infection could be found (eg, labour ward, patient waiting area).
  – *Press releases*. Templates for announcing the objectives of the study and the campaign; countries/facilities were encouraged to engage local media for this purpose.
  – *Social media*. Campaign messaging disseminated and multiplied using social media through HRP's Twitter platform (@HRPresearch).

► *Expand the effect of the campaign*. Countries were encouraged to take ownership over the campaign and develop additional materials and organise activities prior to the start of study data collection.

---

Infections and sepsis remain the major causes of death and disability among women during pregnancy, childbirth, postpartum and post-abortion.[8 9] To respond to this, the Global Maternal and Neonatal Sepsis Initiative was launched in 2016.[4 10] Building on the 2016 SEPSIS-3 definition,[11] the World Health Organization (WHO) led the development of a definition for maternal sepsis as 'a life-threatening condition defined as organ dysfunction resulting from infection during pregnancy, childbirth, postpartum, and post-abortion'.[12] And in 2017, WHO led the Global Maternal Sepsis Study (GLOSS) and Awareness Campaign to assess the burden of maternal infections and sepsis, to validate identification criteria for possible severe maternal infection and maternal sepsis and to raise awareness on maternal sepsis among healthcare providers working in study participating facilities.[13]

Awareness raising has mostly been attempted through campaigns. These have been implemented to increase knowledge, improve attitudes or change behaviours around different health issues.[14–16] Specific to sepsis, the UK Sepsis Trust heads a campaign on sepsis since 2012 and the Global Sepsis Alliance leads efforts aimed at raising sepsis awareness since 2010.[17 18] However, neither of these two large campaigns have been specific to maternal sepsis and to our knowledge neither has been thoroughly evaluated to assess for impact in increasing awareness.

This evaluation looked at recognition and exposure to the GLOSS campaign materials and changes in provider awareness of maternal sepsis after campaign implementation. The latter included changes in knowledge on maternal sepsis and perception of enabling environments for identification and management of maternal sepsis.

## METHODS

The GLOSS campaign was designed to accompany the Global Maternal Sepsis Study with the goal of raising awareness on maternal sepsis among healthcare providers working in participating facilities. Details regarding study protocol, including selection of countries and facilities, were published elsewhere.[13] In short, GLOSS was a facility-based, 1-week inception cohort study which enrolled pregnant or recently pregnant women with suspected or confirmed infection at 713 healthcare facilities in prespecified geographical areas located in 52 low, middle and high-income countries.[13]

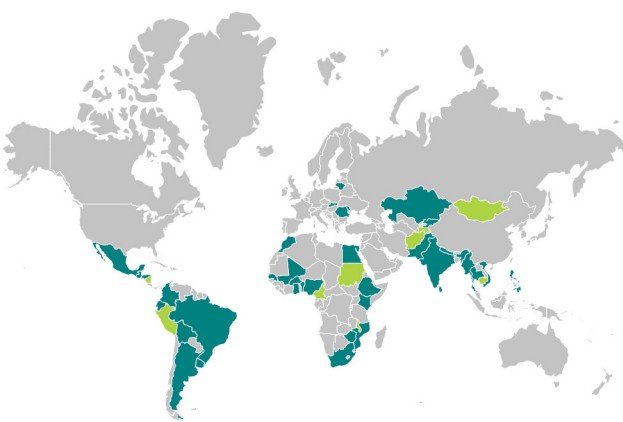

**Figure 1** Countries eligible for the Global Maternal Sepsis Study (GLOSS) '*STOP SEPSIS!*' awareness campaign evaluation (n=46).
Color key: teal: countries included in the GLOSS *STOP SEPSIS! awareness campaign evaluation* (N=37); green: countries eligible for the evaluation but excluded from this analysis because less than 2 responses received (N=9)
The boundaries shown on this map do not imply the expression of any opinion whatsoever on the part of WHO concerning the legal status of any country, or concerning the delimitation of its frontiers or boundaries

### The STOP SEPSIS! awareness campaign

The campaign launch was planned for before study implementation continuing throughout data collection and beyond. It was designed using existing frameworks for public information campaigns, social marketing, health communication and behaviour change.[14 19–21] The development of the campaign included an overarching communication strategy using a multicomponent approach delivering a simple and consistent message through visually attractive media.[22]

The campaign had a soft launch with an online congress on 12 September 2017 and the full campaign roll-out began on 6 November 2017, which included a website, printed materials, social media messaging and press releases. While global coordination of the campaign was undertaken by WHO, implementation of the campaign was the remit of GLOSS country coordinators. Box 1 describes the different actions and components that were necessary for the design and development of the *STOP SEPSIS! awareness campaign.*

### Evaluation of the STOP SEPSIS! awareness campaign

We used an independent sample precampaign/postcampaign design through online and paper-based surveys. Details regarding the definition used for awareness for this campaign, survey formulation and dissemination, including analysis of baseline data, have been published elsewhere.[23] Briefly, a precampaign 32-question survey was developed to gather baseline information on healthcare providers' awareness of maternal sepsis through self-reported knowledge on maternal sepsis and perception of their work environments as enabling for the identification and management of maternal sepsis. Knowledge was assessed through questions relating to

whether respondents had heard of maternal sepsis, correct identification of criteria that define maternal sepsis (infection plus organ dysfunction) and identification of correct initial management of maternal sepsis and infections (antibiotics and fluids) when maternal sepsis was suspected in the case vignette presented in the survey. Perception of enabling environments was assessed

**Table 1** Characteristics of respondents and the facilities in which they work at precampaign and postcampaign surveys (n=2188)

| Respondent characteristics | Precampaign (n=1155) | | Postcampaign (n=1033) | |
|---|---|---|---|---|
| | n | % | n | % |
| Age (years) | 1147 | | 1020 | |
| <31 | 354 | 31 | 301 | 30 |
| 31–40 | 389 | 34 | 407 | 40 |
| >40 | 404 | 35 | 312 | 31 |
| Sex | 1153 | | 1022 | |
| Male | 287 | 25 | 223 | 22 |
| Female | 866 | 75 | 799 | 78 |
| Qualification | 1151 | | 1025 | |
| Nurse/auxiliary nurse/midwife | 440 | 38 | 456 | 44 |
| Physician | 561 | 49 | 456 | 44 |
| Resident | 150 | 13 | 113 | 11 |
| Years of work experience | 1107 | | 970 | |
| <10 | 541 | 49 | 476 | 49 |
| 10–20 | 349 | 32 | 320 | 33 |
| >20 | 217 | 20 | 174 | 18 |
| Region | 1155 | | 1033 | |
| Africa | 224 | 19 | 226 | 22 |
| Asia | 173 | 15 | 170 | 16 |
| Eastern Mediterranean | 171 | 15 | 165 | 16 |
| Europe† | 137 | 12 | 97 | 9 |
| Latin America | 450 | 39 | 375 | 36 |
| Level of the facility in which respondent works | 1153 | | 1033 | |
| I | 127 | 11 | 166 | 16 |
| II | 236 | 20 | 258 | 25 |
| III | 790 | 69 | 609 | 59 |
| Respondent worked in a public facility* | 1154 | | 1033 | |
| Yes | 937 | 81 | 928 | 90 |
| No | 217 | 19 | 105 | 10 |
| Country implemented an expanded version of campaign* | 1155 | | 1033 | |
| Yes | 705 | 39 | 533 | 52 |
| No | 450 | 61 | 500 | 48 |

*P<0.05.
†Includes countries in Central Asia (Kazakhstan, Kyrgyzstan and Tajikistan), in line with WHO regions.

through self-reported confidence in making right decisions, reported availability of resources for correct identification and management, and feeling of support from their work environments in dealing with maternal sepsis, using a 5-point Likert scale. The same survey was administered at postcampaign to assess changes in knowledge and perception of their environments; 14 additional questions were included in the postsurvey which considered respondents' recognition of and exposure to the campaign, such as knowledge about the study and the campaign, message recall, engagement with social media for the campaign and whether the campaign materials prompted changes in behaviour. See online supplementary appendix 1 for a copy of the surveys.

Eligible respondents were healthcare providers working in GLOSS participating facilities in countries that received financial support for campaign implementation (n=46); we excluded all surveys from respondents that did not explicitly state that they were providers caring for women with infections in healthcare facilities (eg, hospital administrators, or community health workers, or if the field was left blank) and from countries with less than two responses at either precampaign or postcampaign (n=9). See figure 1 for a map of all the countries eligible for this evaluation. The surveys were distributed using a snowballing technique and were available in seven languages: Arabic, English, French, Portuguese, Russian, Spanish and Vietnamese. The surveys were available for over 30 days (precampaign between 29 September and 5 November 2017, postcampaign between 31 January and 11 March 2018). Weekly reminders were sent through the online tool and via email to non-respondents. Targeted outreach was undertaken in countries with fewer than two responses. The campaign was active between 6 November 2017 and 15 January 2018; however, countries were encouraged to continue to use the materials beyond GLOSS study implementation.

### Data analysis

We used descriptive analysis to provide frequencies and percentages for the characteristics of the sample, knowledge and perceived enabling environments and for all the questions relating to campaign recognition and exposure. The latter were assessed through postcampaign surveys only and complemented with self-reported accounts by GLOSS country coordinators. Text-based responses were codified into numerical values according to common emerging themes. All Likert scale answers were dichotomised assigning a 1 to the two most favourable responses (ie, they felt *very* or *somewhat* confident about making the right decision) and 0 to the combination of remaining options (*neutral, not very confident* or *not confident at all*). While previously we assessed dichotomisation using 1 to the single most favourable response (ie, respondent felt *very confident*) and a 0 to the combination of remaining options (*somewhat confident, neutral, not very confident* or *not confident at all*)[23] we decided to include a more flexible definition of confidence, perception of availability of resources and feeling of support to allow for a more robust denominator that would enable comparisons.

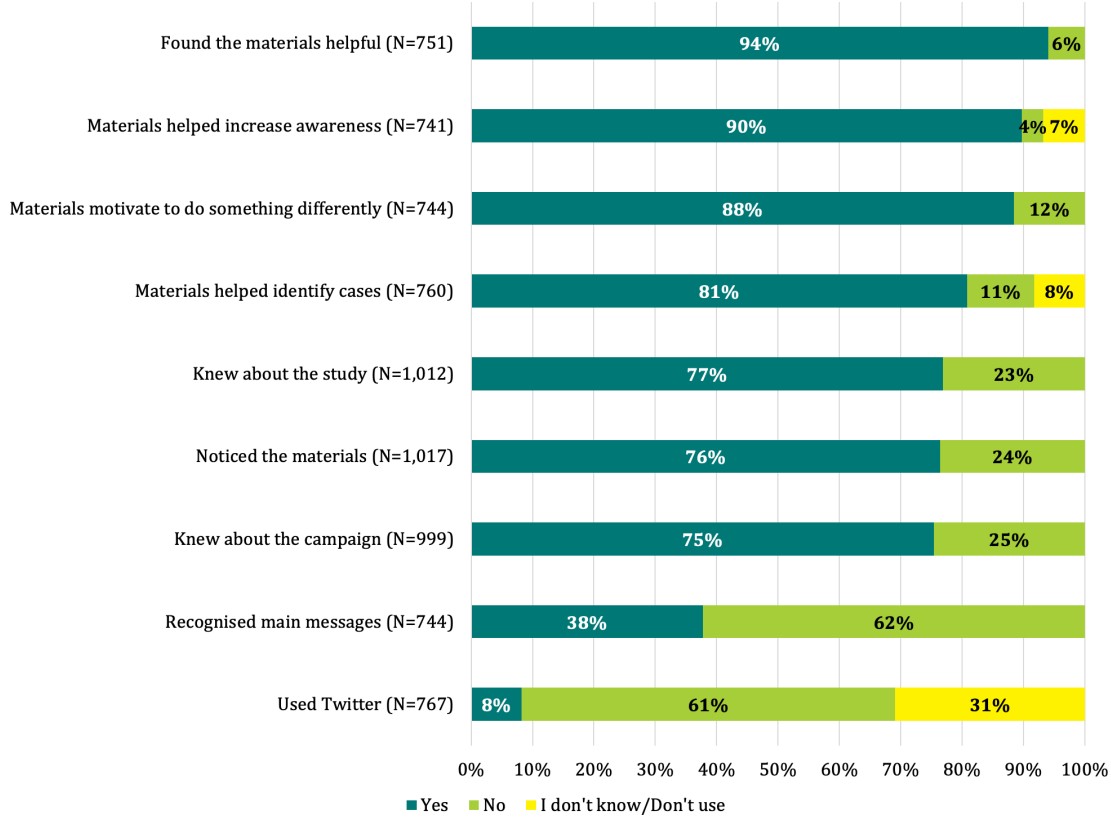

**Figure 2** Measures of campaign exposure in percentages (n=1033).

See online supplementary appendix 2 for results of the overall analysis using this second dichotomisation not used in this evaluation.

To assess the impact of the campaign we conducted several analyses. First, we calculated percentage change ([(% in post − % in pre)/% in pre]×100) and estimated ORs to determine differences in respondent knowledge and perception of enabling environments after campaign implementation relative to baseline measure for the total sample and by respondent characteristics. Due to the methodology used for survey dissemination and anonymity of surveys, this was not a matched sample, paired response precampaign/postcampaign. Sensitivity analyses were conducted restricting the population to facilities from which we received at least one survey response and at least two responses, and to countries with more than 30 survey responses per country at precampaign and at postcampaign.

Second, we used multivariate logistic regression models to explore the association between respondents and facilities' characteristics at precampaign and postcampaign and change in components of awareness after campaign implementation. Based on analysis of baseline data and our assumptions on characteristics that would be associated with levels of awareness,[23] we included the following variables in the model: whether respondent was a physician, years of work experience, region where the respondent worked, whether the country had implemented an expanded version of the campaign and

whether the facility was a level III facility. Countries were considered to have implemented an expanded version of the campaign if they had printed and displayed all posters and infographics, prepared and disseminated press releases and if they had organised other activities or developed other materials for the campaign. Since less than 20% of respondents participated in the World Sepsis Congress Spotlight we did not include this variable in our models. We looked at effect modification by examining interactions between the time of the survey (precampaign or postcampaign) and each of the characteristics included in the model.

We used Pearson's $\chi^2$ test to compare proportions and Wald's test to assess for significant differences in the models including interaction terms. Logistic regression analysis was used to estimate ORs between pre- and post-, crude and adjusted, clustering at the geographical area level. Statistical significance is reported at $p<0.05$. Stata (V.14.2, College Station, TX) was used for the analyses.

### Patient and public involvement

This research was done without patient or public involvement. While the development of the campaign was done with input from study regional and country coordinators, respondents to the surveys were not invited to comment on the study design or to contribute to the writing of this manuscript given their anonymity.

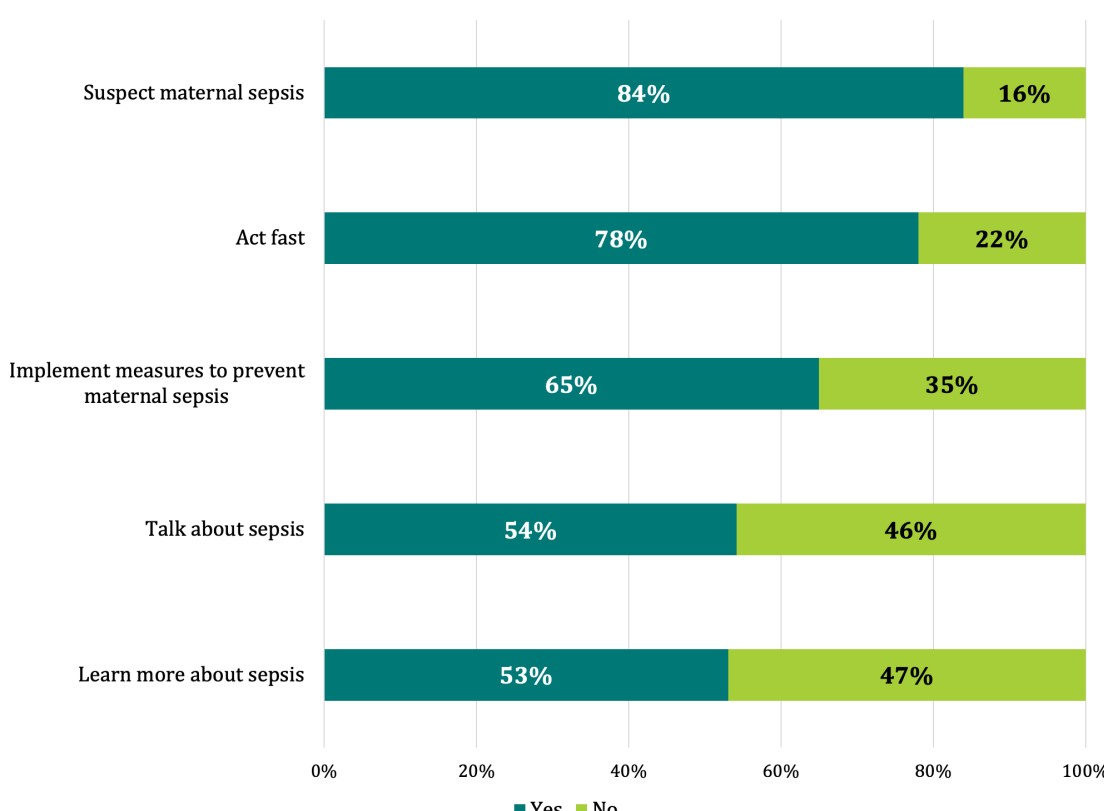

**Figure 3** Responses when answering Yes to the question 'Did the information provided in the materials motivate you to do something differently than before?' (n=658).
(Respondents were able to check as many response options as needed.)

### Box 2  Accounts from the field*

*Implementation of the campaign changed the way the city's providers acted. First, it helped in bridging the gap between academics and providers, which, in turn, helped motivate the entire staff around the study. The campaign helped us all feel more committed with the study. And, most importantly, it helped shed light on a problem (maternal sepsis) that we hadn't made public before.* (Cali, Colombia)

*In (our) facility there was already a protocol for sepsis early recognition, but the campaign, as well as the study made it come alive again. Sepsis was on everyone's eyes and mouths. The teams were very permeable to knowledge and eager to recognize and treat sepsis immediately.* (Campinas, Brazil)

*Participation in the campaign allowed me to see that we can find cases of maternal sepsis in the most diverse locations in a facility. And that invariably the most complex cases were those resulting from a condition that was neglected or treated incorrectly/untimely.* (Maputo, Mozambique)

*Despite having some protocols in place, during the campaign and study we realized that these were not sufficient to detect women with infection. This campaign was very important and helped us find a lot of cases that might have been missed otherwise (…) We are planning on improving reporting mechanisms of any suspected cases and supportive supervision and surveillance as a result of this study.* (Ulaanbaatar, Mongolia)

*As a result of our participation in GLOSS [Global Maternal Sepsis Study], we actually committed as a Program in our 2017 Maternal Death Review Forum to eliminate maternal sepsis as a cause of maternal death.* (Manila, Philippines)

*(Since implementing the GLOSS awareness campaign at a national level, we noticed that) we have prioritized the identification and suspicion of maternal and neonatal sepsis in all level I facilities, in specialized hospital care, and in the public health agenda.* (Mexico City, Mexico)

*These reports first appeared in a blog post on the Merck for Mothers website in April 2018: https://www.msdformothers.com/blog/assessing-addressing-maternal-sepsis.html and in a news story on WHO/HRP's website in September 2018: https://www.who.int/reproductivehealth/maternal-sepsis-mexico/en/

### Role of the funding source

The funders of the study had no role in study design, data collection, data analysis, data interpretation or writing of this original article.

## RESULTS

A total of 2188 surveys met our inclusion criteria. Of these, 1155 from 192 facilities were received at baseline and 1033 from 196 facilities at postcampaign. There were no significant differences in sociodemographic characteristics between respondents at precampaign and postcampaign surveys, except for a higher proportion of respondents working in a public facility at postcampaign and a higher proportion of respondents from countries where an expanded version of the campaign was implemented (table 1). Responses came from the same countries at precampaign and postcampaign. Because of the technique used for survey dissemination and because we did not know the total population of potentially exposed healthcare providers working in GLOSS participating facilities (provider turnover, rotation and replacements

are high), we were unable to calculate a response rate. However, since the campaign was implemented equally at the geographical area level, if providers remained within the study area they would have been exposed to the campaign. Results from the sensitivity analyses showed that overall findings in the subgroups considered were consistent with the results from the complete sample (online supplementary appendix 3); for this reason, we used the entire sample for all subsequent analyses.

We first present the results relating to campaign recognition and exposure and then results relating to changes in knowledge and perception of respondents' work environments.

### Campaign recognition and exposure

Campaign recognition and exposure were high among most of the postcampaign survey respondents. Seventy-six per cent of respondents stated they noticed the materials in their facilities; among those, 94% reported finding the materials helpful, 90% that the materials helped increase awareness on maternal sepsis and 88% that the materials motivated them to do something differently. Only 8% of respondents had used Twitter to amplify the message of the campaign (figure 2). Among respondents who stated that the information provided in the materials motivated them to do something differently than before, 84% stated that it motivated them to suspect maternal sepsis and 78% to act fast (figure 3). Among respondents stating that the materials had not motivated them to do anything differently, 45% said it was because they already knew about maternal sepsis identification and management while 12% stated they had not seen the campaign materials.

Country coordinators shared anecdotal experiences of increased awareness in their facilities and the implementation of changes in practice and policies because of the study and the campaign. These accounts speak of a broader engagement with maternal sepsis identification and management. See box 2 for some examples.

### Knowledge on maternal sepsis and perception of enabling environments

At precampaign survey, 92% of respondents (1049/1144) had heard of maternal sepsis. However, only 16% (109/673) of respondents were able to correctly identify the definition criteria of maternal sepsis and 45% (114/251) identified the correct management for maternal sepsis. In addition, at precampaign, most survey respondents stated that their work environments were enabling for maternal sepsis identification and management: 78% (897/1155) stated that they felt confident of making right decisions, 79% (909/1155) that they perceived resources were available and 80% (921/1155) that they felt supported by their facilities. See table 2 for overall results.

After campaign implementation there was a significant decrease in respondents who stated not having heard of maternal sepsis (−63.4% change; OR 0.35, 95% CI 0.18 to 0.68). There was also a significant increase in perceived confidence in making right decisions with regard to

**Table 2** Respondent knowledge on maternal sepsis and perception of enabling environments for maternal sepsis identification and management at precampaign and postcampaign and changes after campaign implementation (n=2188)

| | Precampaign n/N (%) | Postcampaign n/N (%) | Pre-cOR/post-cOR† (95% CI)‡ | Percentage change |
|---|---|---|---|---|
| Knowledge on maternal sepsis | | | | |
| Had not heard of maternal sepsis§ | 95/1144 (8.3) | 31/1021 (3.0) | 0.35* (0.18 to 0.68) | −63.4 |
| Correctly identified the two criteria to define maternal sepsis¶ | 109/673 (16.2) | 74/647 (11.4) | 0.67 (0.43 to 1.17) | −29.4 |
| Correctly identified management of sepsis when maternal sepsis was suspected** | 114/251 (45.4) | 142/239 (59.4) | 1.76 (0.73 to 4.21) | 30.8 |
| Perception of enabling environment for maternal sepsis identification and management | | | | |
| Confident of making right decisions | 897/1155 (77.7) | 861/1033 (83.4) | 1.44* (1.01 to 2.06) | 7.3 |
| Resources available to make right decisions | 909/1155 (78.7) | 814/1033 (78.8) | 1.01 (0.68 to 1.49) | 0.1 |
| Supported by facility in making right decisions | 921/1155 (79.7) | 840/1033 (81.3) | 1.11 (0.80 to 1.54) | 2.0 |

Percentage change: [(% in post − % in pre)/% in pre]×100.
*P<0.05.
†Refers to OR between precampaign and postcampaign; OR calculated clustering at the geographical area level.
‡Reference group: precampaign.
§Responded No to the question 'Have you ever heard of the term maternal sepsis?'
¶Answered Infection and Organ Dysfunction to the question: 'What two criteria best describe maternal sepsis?'
**Answered Fluids and Antibiotics to the question: 'What would be the first two things a woman should receive?', when the respondent answered Infection/Sepsis to the question: 'What would you first think could be causing her to feel this way?'
cOR, crude OR.

maternal sepsis identification and management (7.3% change; OR 1.44, 95% CI 1.01 to 2.06), although this level was quite high at precampaign (78%). There was a slight increase in respondents' ability to identify the correct management when maternal sepsis was suspected after the implementation of the campaign, but this was not statistically significant (30.8% change; OR 1.76, 95% CI 0.73 to 4.21). See online supplementary appendix 4a,b for these results according to respondent and facility characteristics.

After controlling for respondent and facility characteristics, being a physician, having less than 10 years of experience and working in a level III facility were associated with decreased odds of not having heard of maternal sepsis at precampaign (table 3). Respondents from facilities that had implemented an expanded version of the campaign were more likely to have heard of maternal sepsis and identify the correct management of maternal sepsis at postcampaign. Respondents with less than 10 years of experience were more likely to have heard of maternal sepsis at precampaign, but there were no differences across providers with different years of experience after the campaign.

Physicians were more likely to respond that they felt confident in making the right decisions at postcampaign, while being a physician and having more than 20 years of experience had a significant interaction with time of the survey with regard to perception of availability of resources and support from their facilities. At precampaign and postcampaign, respondents with 20 years or more of experience were more likely to perceive availability of resources for making right decisions and to feel supported by their facilities and these differences between groups were significant after the campaign (table 4). No differences in the perception of enabling environments were seen among respondents from facilities that had implemented an expanded version of the campaign.

## DISCUSSION

To the best of our knowledge, this is the first study to assess impact of an awareness campaign aimed at healthcare providers and implemented at a global level where precampaign and postcampaign data were collected in addition to measures relating to campaign recognition and exposure. Most healthcare providers stated that the campaign helped increase awareness of maternal sepsis and motivated them to do something differently, particularly to suspect maternal sepsis and act faster. Reports from the field also support this finding that exposure to the campaign increased sensitisation to maternal infections and sepsis. Moreover, most survey respondents had heard of maternal sepsis even before campaign implementation; after the campaign this increased significantly. Although most respondents perceived their enabling environments in a positive way before campaign implementation, there was an increase in respondent confidence to make the right decisions regarding maternal sepsis identification and management after campaign implementation.

The *STOP SEPSIS! awareness campaign* implementation was effective with regard to respondents' recognition of and exposure to the campaign; other campaign evaluations have used these measures to positively assess short-term impact of campaigns.[24–26] Furthermore, consistent and repeat exposure to campaign messaging has shown to increase awareness[14]; while exposure was only measured over the course of this evaluation period corresponding to the intended implementation period of the campaign,

**Table 3** Knowledge on maternal sepsis adjusted for respondents' characteristics (n=2188)

| Respondent characteristics | Had not heard about maternal sepsis | | | | | Correctly identified the two criteria to define maternal sepsis | | | | | Correctly identified management of sepsis when maternal sepsis was suspected | | | | |
|---|---|---|---|---|---|---|---|---|---|---|---|---|---|---|---|
| | Precampaign | | Postcampaign | | Wald's test | Precampaign | | Postcampaign | | Wald's test | Precampaign | | Postcampaign | | Wald's test |
| | aOR | 95% CI | aOR | 95% CI | P value | aOR | 95% CI | aOR | 95% CI | P value | aOR | 95% CI | aOR | 95% CI | P value |
| Physician | 0.29* | 0.10 to 0.85 | 0.58 | 0.17 to 1.93 | 0.108 | 1.78 | 0.80 to 3.95 | 3.85* | 1.54 to 9.60 | 0.175 | 2.07* | 1.24 to 3.45 | 2.81* | 1.08 to 7.30 | 0.583 |
| Years of work experience | | | | | 0.035 | | | | | 0.352 | | | | | 0.775 |
| <10 | 0.50* | 0.26 to 0.96 | 1.57 | 0.77 to 3.18 | | 1.08 | 0.60 to 1.96 | 0.86 | 0.55 to 1.34 | | 0.86 | 0.53 to 1.40 | 0.61 | 0.30 to 1.23 | |
| 10–20 | 1 (ref) | | 1 (ref) | | | 1 (ref) | | 1 (ref) | | | 1 (ref) | | 1 (ref) | | |
| >20 | 1.21 | 0.61 to 2.38 | 0.73 | 0.19 to 2.87 | | 1.17 | 0.59 to 2.30 | 1.50* | 1.02 to 2.21 | | 1.59 | 0.60 to 4.22 | 0.9 | 0.30 to 2.71 | |
| Country implemented an expanded version of campaign | 1.54 | 0.46 to 5.12 | 0.21* | 0.06 to 0.78 | 0.004 | 1.18 | 0.53 to 2.65 | 0.86 | 0.31 to 2.34 | 0.281 | 2.78* | 1.01 to 7.59 | 8.02* | 2.03 to 31.73 | 0.437 |
| Respondent worked in a level III facility | 0.45* | 0.21 to 0.96 | 1.79 | 0.59 to 5.42 | 0.006 | 1.63 | 0.77 to 3.45 | 2.80* | 1.32 to 5.94 | 0.314 | 2.10 | 0.85 to 5.16 | 1.14 | 0.43 to 3.02 | 0.406 |

Countries were considered to have implemented an expanded version of the campaign if they had printed and displayed all posters and infographics, prepared and disseminated press releases and if they had organised other activities or developed other materials for the campaign.

Adjusting for whether respondent was a physician, years of work experience, region, whether the country implemented an expanded version of the campaign and whether respondent worked in a level III facility, clustering at the geographical area level.

Wald's test used to assess for differences in the models including interaction terms.

*P<0.05.

aOR, adjusted OR.

**Table 4** Perception of enabling environments for maternal sepsis identification and management adjusted for respondents' characteristics (n=2188)

| Respondent characteristics | Confident of making right decisions | | | | | Resources available to make right decisions | | | | | Supported by facility in making right decisions | | | | |
|---|---|---|---|---|---|---|---|---|---|---|---|---|---|---|---|
| | Precampaign | | Postcampaign | | Wald's test | Precampaign | | Postcampaign | | Wald's test | Precampaign | | Postcampaign | | Wald's test |
| | aOR | 95% CI | aOR | 95% CI | P value | aOR | 95% CI | aOR | 95% CI | P value | aOR | 95% CI | aOR | 95% CI | P value |
| Physician | 1.02 | 0.64 to 1.64 | 1.69* | 1.19 to 2.40 | 0.246 | 0.81 | 0.55 to 1.20 | 1.39 | 0.99 to 1.95 | 0.021 | 0.78 | 0.55 to 1.09 | 1.26 | 0.83 to 1.92 | 0.017 |
| Years of work experience | | | | | | | | | | | | | | | |
| <10 | 0.74 | 0.51 to 1.09 | 0.88 | 0.63 to 1.22 | 0.025 | 0.63* | 0.41 to 0.97 | 0.82 | 0.53 to 1.27 | 0.014 | 0.70 | 0.46 to 1.06 | 0.95 | 0.69 to 1.30 | 0.038 |
| 10–20 | 1 (ref) | | 1 (ref) | | | 1 (ref) | | 1 (ref) | | | 1 (ref) | | 1 (ref) | | |
| >20 | 1.20 | 0.76 to 1.91 | 2.54* | 1.38 to 4.64 | | 1.66* | 1.08 to 2.54 | 2.48* | 1.49 to 4.13 | | 1.60* | 1.07 to 2.40 | 2.78* | 1.84 to 4.20 | |
| Country implemented an expanded version of campaign | 0.65 | 0.36 to 1.16 | 0.91 | 0.56 to 1.48 | 0.605 | 1.00 | 0.52 to 1.94 | 1.57 | 0.85 to 2.88 | 0.241 | 1.32 | 0.86 to 2.03 | 1.37 | 0.90 to 2.07 | 0.092 |
| Respondent worked in a level III facility | 0.90 | 0.55 to 1.47 | 0.64* | 0.41 to 1.00 | 0.581 | 1.39 | 0.86 to 2.23 | 1.27 | 0.79 to 2.04 | 0.255 | 0.88 | 0.53 to 1.46 | 1.18 | 0.67 to 2.07 | 0.116 |

Countries were considered to have implemented an expanded version of the campaign if they had printed and displayed all posters and infographics, prepared and disseminated press releases and if they had organised other activities or developed other materials for the campaign.

Adjusting for whether respondent was a physician, years of work experience, region, whether the country implemented an expanded version of the campaign and whether respondent worked in a level III facility, clustering at the geographical area level.

Wald's test used to assess for differences in the models including interaction terms.

*P<0.05.

aOR, adjusted OR.

the fact that most respondents stated the campaign raised awareness is a promising trend in the right direction.

Overall knowledge about maternal sepsis increased from precampaign to postcampaign implementation among respondents to our survey with regard to having heard about maternal sepsis. Our finding that overall knowledge increased is supported by existing literature that suggests that campaigns can increase knowledge on a specific topic among healthcare providers[15 27 28] as well as among the general population.[24 29 30] The fact that there was a slight increase in identifying the correct management of maternal sepsis is important. Research has shown that knowing what is needed to manage maternal sepsis correctly and early management of maternal sepsis are critical to implementing any changes in providers' behaviour and improving maternal health outcomes.[6 31 32] The low number of providers able to identify the two criteria defining maternal sepsis might be more a reflection of the lack of consensus on this condition prior to 2017, rather than a shortcoming of the campaign.[12] The GLOSS awareness campaign was associated with reducing differences among groups of healthcare providers depending on their qualifications or years of experience. This speaks to the importance of including healthcare providers with different qualifications and years of experience in awareness raising efforts.

We found there were overall increases in respondent confidence in making right decisions about maternal sepsis identification and management, but no significant changes with regard to overall respondent perception of availability of necessary resources and feeling supported by their facilities. Evidence shows that confidence can affect clinical performance[33] and that high levels of confidence among healthcare providers can have a positive impact on patients' perception of experience of care.[34] However, the change in perception of availability of resources and support limited to physicians and more experienced providers raises a broader question on actions that facilities need to take to empower all healthcare workers in feeling that they have the necessary resources and feel supported to provide quality care. This is especially important if we consider that a more restrictive definition of enabling environments results in much lower overall levels of perceived confidence, perception of availability of resources and feeling of support. Perceived lack of availability of resources may also be a product of increased awareness of what is necessary to address maternal sepsis. These findings are a call to hospital administrators and policymakers to foster enabling environments and secure availability and access to life-saving resources.

Sepsis awareness is gaining traction on global agendas[4 10 17]; this is supported by evidence from two studies looking at internet searches on sepsis,[35 36] meaning increases resulting from this campaign could be responding to natural trends or other factors. It is also possible that awareness was raised by having participated in the research study and not necessarily because of the campaign; disentangling the effect of the campaign from that of the implementation of the research study was impossible. Understanding whether any of these changes are sustained over time would provide us with further information on the lasting effects of the campaign.

Literature shows that while a campaign can help in raising awareness, it is insufficient in allowing for changes in behaviour.[14 20] While behaviour change is important in impacting population-level health, it is one of many components needed to make significant improvements; evidence from this study, similar to others, highlights the need for health system improvements such as availability of critical resources and support to improve maternal outcomes.[37] Assessing the impact that increased awareness resulting from a campaign has on behaviour change would provide us with supporting evidence that campaigns can help in improving health outcomes.

This study has some limitations. First, we used a precampaign/postcampaign methodology with no control group which does not allow to discern the impact of the campaign alone. Second, the method used to disseminate the survey and the fact that surveys were anonymous made it impossible to match responses at precampaign and postcampaign. Surveys were anonymous to encourage providers to respond and remove potential response bias. However, it is to note that characteristics of participants at precampaign and postcampaign were similar. Third, because implementation of the campaign was left up to country coordinators, campaign fidelity was only assessed through healthcare provider self-report at postcampaign surveys. Fourth, this evaluation was restricted to the duration of the study follow-up period, hence providing insight into early findings only and limiting our knowledge of lasting impact of the campaign, which was beyond the goal of this activity. However, our findings suggest that campaigns can have at least short-term effects on provider's knowledge and confidence. The positive perception of the campaign materials is encouraging. And fifth, since baseline data were collected after the soft launch of the campaign, the effect of the campaign may have been minimised because awareness had already been increased through exposure to the online congress as well as other global activities on sepsis conducted by other groups. However, we know that less than 20% of respondents participated in the congress.

Our findings have implications for both practice and research. On the one hand, there appear to be benefits to coupling large multicountry studies with awareness campaigns. A campaign targeting healthcare providers can promote their engagement with research studies being conducted, potentially improving study outcomes. There is also evidence that including an awareness campaign creates an environment prime to implementing changes to clinical practice as per research study protocol. On the other hand, there is a clear need for additional research to identify lasting effects of awareness campaigns, especially as global initiatives focus on increasing awareness on maternal health issues.

A campaign designed to raise awareness among healthcare providers working in facilities participating in a global research study was associated with an increase in having heard of maternal sepsis, as well as increased provider

perception of confidence in making correct decisions. Offering healthcare providers with the information to make accurate and timely decisions while promoting environments that enable self-confidence and support could improve maternal sepsis identification and management, which can ultimately have an impact on maternal health outcomes.

**Author affiliations**
[1]UNDP/UNFPA/UNICEF/WHO/World Bank Special Programme of Research, Development and Research Training in Human Reproduction (HRP), Department of Sexual and Reproductive Health and Research, World Health Organization, Geneva, Switzerland
[2]Department of Social Medicine, Ribeirão Preto Medical School, Ribeirão Preto, Brazil
[3]Centro Rosarino de Estudios Perinatales (CREP), Rosario, Argentina
[4]Research Institute of Health Sciences, Ouagadougou, Burkina Faso
[5]Department of Obstetrics and Gynaecology, University of Ibadan College of Medicine, Ibadan, Oyo, Nigeria
[6]National Perinatal Epidemiology Unit, University of Oxford, Oxford, UK
[7]Obstetrics and Gynecology Faculty of Medicine, Khon Kaen University, Khon Kaen, Thailand
[8]Department of Obstetrics and Gynaecology, Lithuanian University of Health Sciences, Kaunas, Lithuania
[9]Department of Obstetrics and Gynaecology, Ain Shams University, Cairo, Egypt
[10]Department of Obstetrics/Gynaecology, Eduardo Mondlane University, Maputo, Mozambique
[11]Department of Obstetrics and Gynecology, University of Nairobi, Nairobi, Kenya

**Acknowledgements** The authors thank Khalid Yunis for his contributions to the development of the campaign, Soe Soe Thwin and Cristina Cuesta for their assistance in the planning and review of statistical analyses and all the respondents to the online and paper-based survey. The authors also acknowledge the contribution and lifelong achievements of coauthor Bukola Fawole, who passed away prior to the publication of this article.

**Collaborators** WHO Global Maternal Sepsis Study Research Group (in alphabetical order): *Afghanistan*: Mohammad Iqbal Aman, Bashir Noormal. *Argentina*: Virginia Díaz, Marisa Espinoza, Julia Pasquale. *Belgium*: Charlotte Leroy, Kristien Roelens, Griet Vandenberghe. *Benin*: M Christian Urlyss Agossou, Sourou Goufodji Keke, Christiane Tshabu Aguemon. *Bolivia*: Patricia Soledad Apaza Peralta, Víctor Conde Altamirano, Rosalinda Hernández Muñoz. *Brazil*: José Guilherme Cecatti, Carolina Ribeiro do Valle. *Burkina Faso*: Vincent Batiene, Kadari Cisse, Henri Gautier Ouedraogo. *Cambodia*: Cheang Kannitha, Lam Phirun, Tung Rathavy. *Cameroon*: Elie Simo, Pierre-Marie Tebeu, Emah Irene Yakana. *Colombia*: Javier Carvajal, María Fernanda Escobar, Paula Fernández. *Denmark*: Lotte Berdiin Colmorn, Jens Langhoff-Roos. *Ecuador*: Wilson Mereci, Paola Vélez. *Egypt*: Yasser Salah Eldin, Alaa Sultan. *Ethiopia*: Abdulfetah Abdulkadir Abdosh, Alula M Teklu, Dawit Worku. *Ghana*: Richard Adanu, Philip Govule, Charles Noora Lwanga. *Guatemala*: William Enrique Arriaga Romero, María Guadalupe Flores Aceituno. *Honduras*: Carolina Bustillo, Rigoberto Castro, Bredy Lara. *India*: Vijay Kumar, Vanita Suri, Sonia Trikha. *Italy*: Irene Cetin, Serena Donati, Carlo Personeni. *Kazakhstan*: Guldana Baimussanova, Saule Kabylova, Balgyn Sagyndykova. *Kenya*: George Gwako, Alfred Osoti, Zahida Qureshi. *Kyrgyzstan*: Raisa Asylbasheva, Aigul Boobekova, Damira Seksenbaeva. *Lebanon*: Faysal El Kak, Saad Eddine Itani, Sabina Abou Malham. *Lithuania*: Meilė Minkauskienė, Diana Ramašauskaitė. *Malawi*: Owen Chikhwaza, Luis Gadama, Eddie Malunga. *Mali*: Haoua Dembele, Hamadoun Sangho, Fanta Eliane Zerbo. *Mexico*: Filiberto Dávila Serapio, Nazarea Herrera Maldonado, Juan I Islas Castañeda. *Republic of Moldova*: Tatiana Cauaus, Ala Curteanu, Victor Petrov. *Mongolia*: Yadamsuren Buyanjargal, Seded Khishgee, Bat-Erdene Lkhagvasuren. *Morocco*: Bouchra Assarag, Amina Essolbi, Rachid Moulki. *Mozambique*: Zara Jaze, Arlete Mariano, Nafissa Bique Osman. *Myanmar*: Hla Mya Thway Einda, Thae Maung Maung, Khaing Nwe Tin. *Nepal*: Tara Gurung, Amir Babu Shrestha, Sangeeta Shrestha. *Netherlands*: Kitty Bloemenkamp, Marcus J Rijken, Thomas Van Den Akker. *Nicaragua*: María Esther Estrada, Néstor J Pavón Gómez. *Nigeria*: Olubukola Adesina, Chris Aimakhu, Bukola Fawole. *Pakistan*: Rizwana Chaudhri, Saima Hamid, M Adnan Khan. *Peru*: María del Pilar Huatuco Hernández, Nelly M Zavaleta Pimentel. *Philippines*: Maria Lu Andal, Zenaida Dy Recidoro, Carolina Paula Martin. *Romania*: Mihaela Budianu, Lucian Puşcaşiu. *Senegal*: Léopold Diouf, Dembo Guirassy, Philippe Marc Moreira. *Slovakia*: Miroslav Borovsky, Ladislav Kovac, Alexandra Kristufkova. *South Africa*: Sylvia Cebekhulu, Laura Cornelissen, Priya Soma-Pillay. *Spain*: Vicenç Cararach, Marta López, María José Vidal Benedé. *Sri Lanka*: Hemali Jayakody, Kapila Jayaratne, Dhammica Rowel. *Sudan*: Mohamed Elsheikh, Wisal Nabag, Sara Omer. *Tajikistan*: Victoria Tsoy, Urunbish Uzakova, Dilrabo Yunusova. *Thailand*: Thitiporn Siriwachirachai, Thumwadee Tangsiriwatthana. *United Kingdom*: Catherine Dunlop, Marian Knight, David Lissauer. *Uruguay*: Aquilino M Pérez, Jhon Roman, Gerardo Vitureira. *Vietnam*: Dinh Anh Tuan, Luong Ngoc Truong, Nghiem Thi Xuan Hanh. *Zimbabwe*: Mugove Madziyire, Thulani Magwali, Stephen Munjanja.Regional coordinators: Edgardo Abalos, Adama Baguiya, Mónica Chamillard, Bukola Fawole, Marian Knight, Seni Kouanda, Pisake Lumbiganon, Ashraf Nabhan, Ruta Nadisauskiene. Technical Advisory Group: Linda Bartlett, Fernando Bellissimo-Rodrigues, Shevin T Jacob, Sadia Shakoor, Khalid Yunis. Data management and analysis: Liana Campodónico, Cristina Cuesta, Hugo Gamerro, Daniel Giordano. WHO coordinating unit: Fernando Althabe, Mercedes Bonet, Vanessa Brizuela, A Metin Gülmezoglu, João Paulo Souza.

**Contributors** The evaluation plan and first draft of this original article was conceived by VB. MB provided substantial contributions to the data planning and analysis and first versions of this manuscript. The idea for the inclusion of an awareness campaign was first conceived by JPS. CLTR, JPS and EA provided additional feedback on first draft and data analysis. VB, MB, JPS, EA, AB, BF, MK, PL, MM, AN, NBO and ZPQ were instrumental to campaign design and implementation in their regions/countries and provided input to data collection tools used for this evaluation. All authors read and approved the final version. The WHO GLOSS Research Group contributed to different aspects of the study, from protocol development to acquisition of data and coordination of study in their countries. VB, MB and CLTR had full access to all the data and VB has final responsibility for the decision to submit for publication.

**Funding** This work was supported by the UNDP/UNFPA/UNICEF/WHO/World Bank Special Programme of Research, Development and Research Training in Human Reproduction (HRP), Department of Sexual and Reproductive Health and Research, WHO, Geneva, Switzerland (project A65787), Merck Sharp & Dohme, a wholly owned subsidiary of Merck and Co (Kenilworth, NJ, USA), through its Merck for Mothers programme, and the US Agency for International Development (USAID; grant number GHA-G-00-09-00003).

**Disclaimer** The views of the funding bodies have not influenced the content of this manuscript. This article represents the views of the named authors only and does not represent the views of the WHO.

**Map disclaimer** The depiction of boundaries on the map(s) in this article does not imply the expression of any opinion whatsoever on the part of BMJ (or any member of its group) concerning the legal status of any country, territory, jurisdiction or area or of its authorities. The map(s) are provided without any warranty of any kind, either express or implied.

**Competing interests** All authors declare support from WHO/SHR, Merck for Mothers and USAID for the submitted work. VB, MB and JPS were employed by WHO at the time of the study. No other relationships or activities could appear to have influenced the submitted work.

**Patient and public involvement** Patients and/or the public were not involved in the design, or conduct, or reporting, or dissemination plans of this research.

**Patient consent for publication** Not required.

**Ethics approval** Ethical approval for the entire study (GLOSS), including the awareness campaign, was obtained from WHO's Ethics Review Committee in Geneva, Switzerland (protocol ID A65787) for a multisite, multicountry project, and from local and facility ethics committees as necessary. A legend was included in the survey stating that response to the survey implied consent to participate; respondents could recuse themselves at any point during the survey, including skipping and not answering certain questions.

**Provenance and peer review** Not commissioned; externally peer reviewed.

**Data availability statement** Data are available upon reasonable request. All enquiries regarding the data and analyses can be made to the corresponding author.

**ORCID iDs**
Vanessa Brizuela http://orcid.org/0000-0002-4860-0828
Mercedes Bonet http://orcid.org/0000-0001-9015-1635
Carla Lionela Trigo Romero http://orcid.org/0000-0002-8470-5272
Edgardo Abalos http://orcid.org/0000-0001-6653-429X
Adama Baguiya http://orcid.org/0000-0003-1016-4896
Bukola Fawole http://orcid.org/0000-0002-2131-539X
Marian Knight http://orcid.org/0000-0002-1984-4575
Pisake Lumbiganon http://orcid.org/0000-0001-9372-0071
Meilė Minkauskienė http://orcid.org/0000-0003-1056-0522
Ashraf Nabhan http://orcid.org/0000-0003-4572-2210
Nafissa Bique Osman http://orcid.org/0000-0001-5832-1630
Zahida P Qureshi http://orcid.org/0000-0003-1764-3136
João Paulo Souza http://orcid.org/0000-0002-2288-4244

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
