## [Reviewer comments · BMJ Open]

ARTICLE DETAILS

TITLE (PROVISIONAL)	Early evaluation of the 'STOP SEPSIS!' WHO Global Maternal Sepsis Awareness Campaign implemented for healthcare providers in 46 low- middle- and high-income countries
AUTHORS	Brizuela, Vanessa; Bonet, Mercedes; Trigo Romero, Carla Lionela; Abalos, E; Baguiya, Adama; Fawole, Adeniran O.; Knight, Marian; Lumbiganon, Pisake; Minkauskienė, Meilė; Nabhan, Ashraf; Osman, Nafissa; Qureshi, Zahida; Souza, Joao Paulo

VERSION 1 – REVIEW

REVIEWER	Matthew Hensley University of Michigan, Pulmonary & Critical Care Medicine
REVIEW RETURNED	07-Jan-2020

GENERAL COMMENTS	Brizuela, Bonet and colleagues have conducted a thoughtful pre-post survey analysis of how awareness campaigns change perceptions and knowledge of maternal sepsis on a global scale. I am impressed by their thorough approach, assessment of a variety of providers, countries, and healthcare settings, analysis, as well as the excellent portrayal of the results. I have only one minor revision to consider: Figure 1, Page 32; Lines 33-34 - Is there any information on why countries in yellow did not implement the campaign (mostly European countries)? Would suggest including this as a limitation, explaining how you think their participation in GLOSS but lack of implementation affected the overall results, if any.
---

REVIEWER	Professor Michael Turner UCD Centre for Human Reproduction, University College Dublin
REVIEW RETURNED	16-Jan-2020

GENERAL COMMENTS	Suggest stating in the abstract that the pre-campaign survey was completed in all cases by at least 23 days before the campaign and that the post campaign survey was completed at least 16 days afterwards I suggest qualifying the word evaluation in the title by "Early" The first paragraph of the Introduction should make clear that the Surviving Sepsis Campaign since it started in 1992 excluded maternal sepsis. The focus on maternal sepsis is only recent. In Table 1 clarify what comparison the p value refers to.
---

	The fact that 8.3% had not heard of maternal sepsis pre-campaign is disturbing. The low number that identified the two criteria to define maternal sepsis is not surprising given the lack of consensus about defining maternal sepsis before 2017. It is disappointing that the campaign made no difference to the resources available. What were the deficiencies in the 21% before and after in Table 21? In Table 3 why use 10-20 years as the reference group? Discussion should highlight the early evaluation of the campaign but the sustainability of improvement needs further study Interesting lack of resources bigger problem in Eastern Mediterranean than Africa Was health literacy considered for the campaign?
--	--

REVIEWER	Rodolfo Gomez Ponce de leon Pan American Health Organization, CLAP WRH
REVIEW RETURNED	17-Mar-2020

GENERAL COMMENTS	congratulations , the report is very valuable for the international
---

VERSION 1 – AUTHOR RESPONSE

Reviewer 1

1- Brizuela, Bonet and colleagues have conducted a thoughtful pre-post survey analysis of how awareness campaigns change perceptions and knowledge of maternal sepsis on a global scale. I am impressed by their thorough approach, assessment of a variety of providers, countries, and healthcare settings, analysis, as well as the excellent portrayal of the results. I have only one minor revision to consider:

Figure 1, Page 32; Lines 33-34 - Is there any information on why countries in yellow did not implement the campaign (mostly European countries)? Would suggest including this as a limitation, explaining how you think their participation in GLOSS but lack of implementation affected the overall results, if any.

AUTHORS' RESPONSE: Many thanks for your words and your comment. As stated in the manuscript on page 9, lines 124-125: "Eligible respondents were healthcare providers working in GLOSS participating facilities in countries that received financial support for campaign implementation (N=46)." Western European countries did not receive funding for the study or the campaign, and in addition, countries like the UK were also implementing nationwide sepsis campaigns (UK Sepsis Trust) meaning they did not implement a separate GLOSS campaign. Lastly, although Western European countries participated in the overall GLOSS study, they did so by implementing an adapted protocol (Bonet et al, Reproductive Health 2018; Bonet et al, Lancet Global Health, 2020 (in press)); we do not think this is a limitation to our study as the application of the GLOSS protocol was distinctly different for Western European countries by design.

To simplify and make the figure more specific to this evaluation, we have amended the figure changing the title to: "Countries eligible for the GLOSS 'STOP SEPSIS!' awareness campaign evaluation (N=46)"

And then showing two categories: (i) countries included in the evaluation (N=37), and (ii) countries eligible but from which ≤ 1 responses were received for the evaluation (hence, not included in the evaluation) (N=9).

We have also modified the text that references the figure in the manuscript to (page 9, line 129): "See Figure 1 for a map of all the countries eligible for this evaluation."

Reviewer 2

1- Suggest stating in the abstract that the pre-campaign survey was completed in all cases by at least 23 days before the campaign and that the post campaign survey was completed at least 16 days afterwards

AUTHORS' RESPONSE: Thanks for the suggestion. We have included some language around survey availability in the abstract –the pre-campaign survey was available for 37 days up until the day before the campaign was launched. The revised text, page 3, lines 4-5 now reads:

"Independent sample pre-post intervention through online and paper-based surveys available for over 30 days before campaign rollout (pre) and after study data collection (post)."

2- I suggest qualifying the word evaluation in the title by "Early"

AUTHORS' RESPONSE: Thanks for this recommendation. As per suggestion by the journal editor, the title has now been changed to:

"Early evaluation of the 'STOP SEPSIS!' WHO Global Maternal Sepsis Awareness Campaign implemented for healthcare providers in 46 low- middle- and high-income countries"

3- The first paragraph of the Introduction should make clear that the Surviving Sepsis Campaign since it started in 1992 excluded maternal sepsis. The focus on maternal sepsis is only recent.

AUTHORS' RESPONSE: We have included some additional text to this end. The sentence now reads (page 5, lines 56-59):

"With updates in 2017 and 2018, the Surviving Sepsis Campaign has been developing guidelines for management and recommended bundles of care for sepsis among adult populations, not specific to pregnant or recently pregnant women, since 2002."

4- In Table 1 clarify what comparison the p value refers to.

AUTHORS' RESPONSE: Thanks for this comment; we believe the way that the table is presented and the accompanying text in the manuscript clarifies that the p value is intended to compare the sample at pre- and post-campaign. On page 12, lines 194-198 we state: "There were no significant differences in sociodemographic characteristics between respondents at pre- and post-campaign surveys, except for a higher proportion of respondents working in a public facility at post-campaign and the higher proportion of respondents from countries where an expanded version of the campaign was implemented (Table 1)."

5- The fact that 8.3% had not heard of maternal sepsis pre-campaign is disturbing.

AUTHORS' RESPONSE: Indeed. But it is promising that after a short campaign, this improved!

6- The low number that identified the two criteria to define maternal sepsis is not surprising given the lack of consensus about defining maternal sepsis before 2017.

AUTHORS' RESPONSE: Exactly. We have added some text in the discussion section to clarify this, including a reference to the statement on maternal sepsis that speaks to the consensus. The added sentence on page 21, lines 289-291 now reads:

"The low number of providers able to identify the two criteria defining maternal sepsis might be more a reflection of the lack of consensus on this condition prior to 2017, rather than a shortcoming of the campaign."

7- It is disappointing that the campaign made no difference to the resources available. What were the deficiencies in the 21% before and after in Table 21?

AUTHORS' RESPONSE: We agree with this comment and have addressed this shortcoming in the manuscript on page 22, lines 319-322: "While behaviour change is important in impacting population-level health, it is one of many components needed to make significant improvements; evidence from this study, similar to others, highlight the need for health systems improvements such as availability of critical resources and support to improve maternal outcomes."

We did not collect information on the specific resources that were perceived as unavailable.

8- In Table 3 why use 10-20 years as the reference group?

AUTHORS' RESPONSE: Using this age group as the reference group allowed us to make comparisons between the very inexperienced and the more experienced.

9- Discussion should highlight the early evaluation of the campaign but the sustainability of improvement needs further study

AUTHORS' RESPONSE: We have mentioned this in our limitations section but have added some text for further clarity. Page 22, lines 332-334 now reads:

"Fourth, this evaluation was restricted to the duration of the study follow-up period, hence providing insight into early findings only and limiting our knowledge of lasting impact of the campaign, which was beyond the goal of this activity."

10- Interesting lack of resources bigger problem in Eastern Mediterranean than Africa

AUTHORS' RESPONSE: Indeed. To clarify this, we have added some text into the manuscript on page 21, lines 305-307 that reads:

"Perceived lack of availability of resources may also be a product of increased awareness of what is necessary to address maternal sepsis".

11- Was health literacy considered for the campaign?

AUTHORS' RESPONSE: The text included in the posters, infographics, and fact sheets was simple and straightforward. Since the campaign was directed at healthcare workers, basic reading and comprehension skills were assumed. In addition, all materials were translated into eight languages allowing for a wider scope of understanding. Lastly, our post-campaign survey asked for specific difficulties in understanding the content of the materials (appendix 1, page 8, question 23.b: "the message was confusing, they were difficult to understand"). The responses to that question were negligible, hence not presented in the results. We are confident that health literacy was not an issue with the intended audience.

Reviewer 3

1- Congratulations, the report is very valuable for the international

AUTHORS' RESPONSE: Thank you for your comment. We also believe that these results are very valuable.

VERSION 2 – REVIEW

REVIEWER	Matthew Hensley University of Michigan, United States of America
REVIEW RETURNED	02-Apr-2020

GENERAL COMMENTS	The authors have answered questions sufficiently and have significantly contributed to field of maternal sepsis research.
---

REVIEWER	Professor Michael Turner UCD Centre for Human Reproduction, University College Dublin
REVIEW RETURNED	06-Apr-2020

GENERAL COMMENTS	I am happy for this paper to be accepted without any further revision
---